# Resolution Adaptive Networks for Efficient Inference

## Reproducibility Summary

*In this paper the authors (Yang et al.) propose a Resolution Adaptive CNN network (RANet) to efficiently classify images at different resolutions depending based on the classification difficulty (prediction confidence). The main idea is that some images are easy to classify, only requiring low resolution and simple features (low computational cost) while other images require higher resolution and larger networks (large computational cost). RANet attempts to classify a low-resolution representation of the image and if the classification confidence is below some threshold, will attempt will try to classify the image with higher and higher resolution with subsequent subnetworks.*

*The authors make two main assertions that we (The Einsteins) hope to verify in this reproducibility study. Firstly, does the RANet architecture offer significant reductions in computational cost, while maintaining accuracy to comparable models. Secondly, in an arbitrary dataset there's such a distribution of 'easy' and 'hard' classifications tasks that would warrant the use of and adaptive networks. We tested the first assumption by creating our implementation of RANet in Tensorflow Keras and found it to have comparable accuracy with VGG16 and Resnet50, while having a reduction in computational time by up to 64%. We tested the second assumption by creating our own "spalling" classifier using the authors' provided code and kept track of how often each subnetwork returned the prediction. We found that the assertion that some images are 'easier' or more 'difficult' to classify is valid, with 74% of spalling images being considered 'hard' to classify and the rest of the spalling images being 'easy' to classify.*

*While, we have not reproduced the authors' results directly, we show that the proposed method shows promise and may have application for real-time and/or AI at the edge applications.*

## 1 Introduction

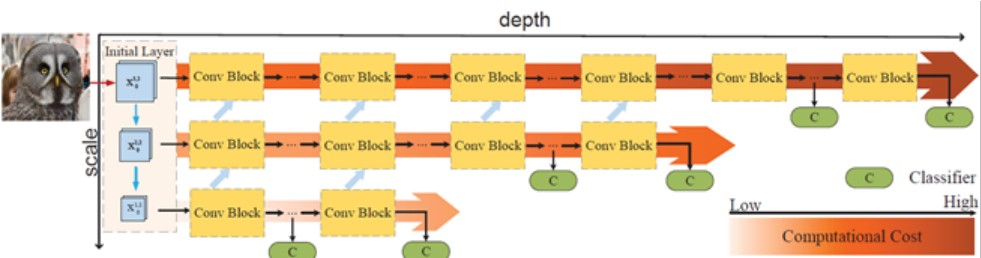

Figure 1: The illustration of an RANet with three scales. Classifiers only operate on feature maps at the lowest resolution

Figure 1, from Yang et al. Provides a good overview of the proposed RANet architecture. To start the input image is resized into 3 scales, where each scale is half of the previous. In a forward pass, the smallest image (scale 1) is passed into the first subnetwork. If the classification confidence of the first subnet is below some threshold, the second subnetwork at scale 2 will be used for classification. The output of the Conv Block at the scale minus one is use 'fused' with the feature maps of the Conv Block of the current scale. This process is repeated until a prediction of high confidence is achieved.

The typical Conv Block in RANet consists of a convolution layer (CONV2D), batch-normalization (BN) and rectified linear (ReLU) activation. To combine the output of the Conv Block at scale and scale minus one, the feature maps of both are concatenated, but the shapes of these outputs must match. The authors propose 2 different methods: up sample feature maps of scale minus one or down sample the feature maps at scale before concatenation.

RANet is trained from end to end using the categorical cross-entropy metric. There are many intermediary classifiers in RANet, this is used in training where the final loss is a weighted sum of these intermediary losses. RANet is trained using stochastic gradient decent (SGD), for 300 epochs, with varying learning rate, momentum of 0.9 and weights decomposition. The dataset is augmented by pixel standardization and random horizontal flips.

## 2    Scope of Reproducibility

The authors assert that some images are easier to classify than others, such that an adaptive network during inference can decrease computational cost by using less computation for easily classified samples. The authors claim that their RANet maintains the same classification accuracy while reducing computation cost of 65%, 56% and 44% compared to GoogLeNet, ResNet and DenseNet respectively. We assert that for some hardware the computational costs are linearly related to computational time. We will implement our own RANet with Tensorflow Keras and compare the computation time with popular CNN models. The authors will train our own networks from scratch and will not train by using Imagenet dataset. For this reason, we will also refrain from training large models due to time and computational costs. Due to the difficulty of working with large datasets, we will also restrict our study to the CIFAR10 dataset. In this study we will compare our RANet to VGG16 and Resnet50, where we expect a roughly 2x decrease in computational time for no loss in classification accuracy.

Additionally, to test the assertion that in a dataset there are images that are easier to classify than others, we will modify one of our datasets to detect concrete spalling to train a binary classifier using the authors' code. We will implement four subnetworks to determine what percentage of final predictions are produced by each subnetwork. To validate this, the authors hope to observe that a significant portion of images are being predicted using the earlier, less computationally intensive subnetworks, proving that spalling images are 'easy' to classify and that RANet can be used to improve the efficiency of existing spalling classifiers.

## 3    Methodology

For the reproducibility project we implemented the proposed RANet architecture with Tensorflow Keras. The original authors had implemented their code in Pytorch, from which we based our layer initializations, but we based our architecture entirely on what was described in the authors' paper. The result is that our implementation and the authors are superficially different but theoretically functions the same. Our RANet model was trained from scratch using CIFAR10 dataset with 80/20 train-test split. We used ImageDataGenerator for image augmentation, learning rate schedule and SGD. Our model achieved a peak validation accuracy of 87% during training, higher accuracies were possible but were not attempted due to time constraints. During training we recorded loss from each subnetwork, and they are combined linearly to calculate the total losses, they are provided in figure 2 and table 1.

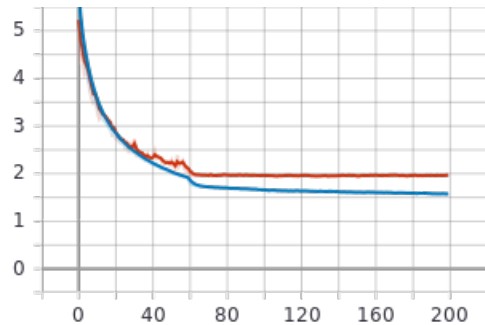

Figure 2: Total Epoch Loss

| | Epoch Loss | Epoch Accuracy |
|---|---|---|
| Small | 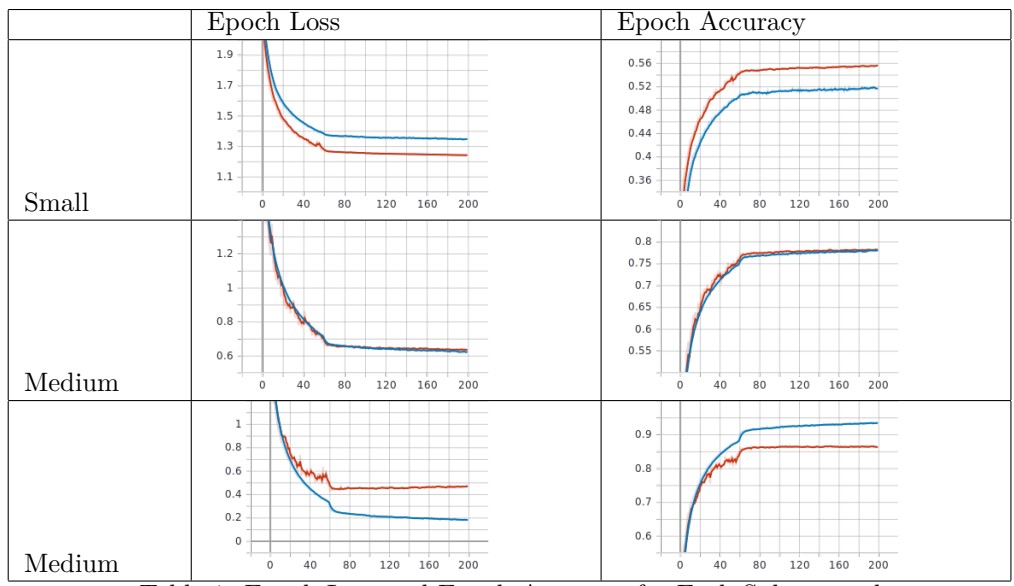 | |
| Medium | | |
| Medium | | |

Table 1: Epoch Loss and Epoch Accuracy for Each Sub-network

From the figures above we observe that the small sub-model achieved an accuracy of 52%, the medium sub-model achieved an accuracy of 78% and finally the large sub-model achieved an accuracy of 87%. These results match our intuition that a reduction in image resolution will result in reduced accuracy. As stated in the scope of reproducibility, we will not be conducting budgeted inference, rather we are interested in average computation time per image. We trained VGG16 and Resnet50 on CIFAR10 to baseline against our RANet model. After a model was trained each was run on a similar experiment pipeline. We recorded the average time for a given model to predict 100 randomly selected validation images, this was done three times per model. Since the speed of RANet depends on the accuracy threshold, the accuracy of the prediction is also recorded, while we assumed VGG16 and Resnet50 will average to the validation accuracy.

For the second experiment, the dataset was provided by the CVISS (Computer Vision and Smart Structures) Lab in the department of Civil and Environmental Engineering at the University of Waterloo. The dataset consists of approximately 1000 earthquake images where spalled concrete damage was labeled with bounding boxes. Using a 80/20 split, we divided the images into training and validation datasets. We then extracted multiple square patches from each image where spalling damage may or may not be present. In total, we extracted approximately 20000 patches for training and 5000 patches for validation of the model.

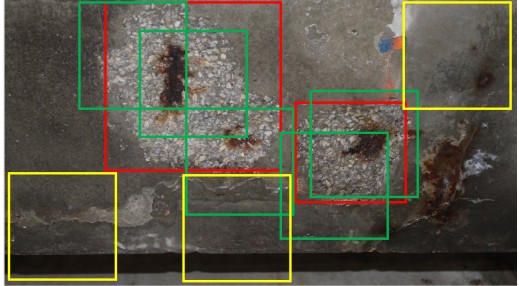

Figure 3: Extracted patches from spalling dataset image. 'Red' - labelled spalling region, 'Green' - spalling extracted patches, 'yellow' - non-spalling extracted patches.

In figure 3, the 'red' annotations are the spalling labels, 'green' are the extracted spalling patches and 'yellow' are non-spalling patches. These image patches are resized to 224 x 224 (ImageNet resolution) and are used for training of the binary classifier. We implemented the author's code and used four subnetworks for classification. We tested our hypothesis on 812 images, and we kept track of the predictions produced by each subnetwork and which subnetwork was utilized for the final prediction.

## 4 Results

We found that RANet decreased computational cost as claimed in the original paper. With no decrease in accuracy RANet was 2x and 2.8x faster than VGG16 and Resnet50.

| Model Architecture | Accuracy | Average Inference Time (ms) |
|---|---|---|
| VGG16 | 86% | 33.1 |
| Resnet50 | 88% | 44.2 |
| RANet | 87% | 16.3 |

Table 2: Experimental Results

The metrics that we used differ from those the authors used in the paper, however our results do support the author's conclusions about the potential such an architecture has reduce the computational cost of inference. For example, we observed a 64% decrease in computational time compared to Resnet50, which is not far from the 56% decrease in computation cost for Resnet reported by the original authors.

The proposed adaptive architecture yielded promising results, we also wanted to test how applicable this would be to a real-world problem in the domain of civil engineering such as spalling classification. We adapted the authors' code and used four subnetworks labelled 'conv_net1', 'conv_net2', 'conv_net3', and 'conv_net4', which scale up based on complexity and size of each subnetwork. Of the 812 images that were classified using RANet, 604 utilized the largest network 'conv_net4', 157 images utilized 'conv_net3', 41 images utilized 'conv_net2', and only 10 images utilized the smallest 'conv_net1' network.

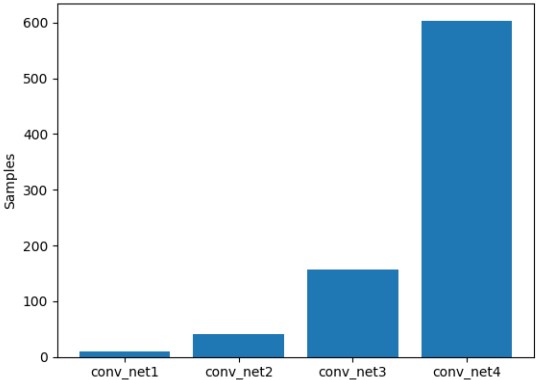

Figure 4: Number of image samples classified using each subnetwork of RANet.

We found that there was indeed a distribution of 'easy' and 'difficult' to classify images. However, 74% of spalling images can be considered 'hard' to classify and required 'conv_net4', and the rest of the spalling images are 'easy' to classify. This proves that spalling images require a high computational cost to classify and might not necessarily benefit from implementing RANet because we would still be relying on large networks that are just as computationally intensive as traditional CNN classification techniques.

**What was easy**

The theory behind the proposed adaptive architecture the authors describe is straight forward and intuitive to understand. The concept can be easily adapted for various and diverse applications. The authors' choice to use standard datasets also made reproducibility easier.

**What was difficult**

A difficult task in this project is to code RANet in Tensorflow Keras, due to the differing structure of the Keras model class. We found very few resources on how to make an adaptive model in Tensorflow and Keras documentation and tutorials. Additionally, it was difficult to exactly replicate the author's RANet as their layer's parameters were not provided. This also resulted in difficulties during training as we tried to achieve higher validation accuracies.

The author's code is heavily borrowed from previous work and there were very few comments nor direction. The author's code was easy to use and it was easy to implement our custom spalling dataset. However, we found it difficult to modify the author's code to produce Figure 4, the number of image samples classified using each subnetwork of RANet.

Finally, due to the university closure and that our team members operated in different time zones, it caused delays and difficulties in communication. With much hard work we were able to produce this deliverable, while may not be to the standard of this journal, the journey here has been very rewarding. We should like to thank Dr. Chul Min Yeum for access to the dataset and Juan Park for hosting our test machines as we leached many kilowatt hours from his meter.

**Communication with original authors**

We had no contact with the original paper's authors.

**Open-Source Repository**

We published our TF2 implementation in a GitHub repository: https://github.com/MACILLAS/syde671.git

