# OpenReview forum: "Resolution Adaptive Networks for Efficient Inference"
_ML_Reproducibility_Challenge/2020 — Reject_

### Official Review · AnonReviewer2 · 2021-03-01
**A complementary study of the original paper**

**Rating:** 7
**Confidence:** 4

**Review:**

Instead of reproducing exact results from the original paper, the submission implemented their own version and verified that the assumptions and results of the original paper do hold. In general, this is a good complementary study for the original paper.

The subject of study in the original paper is a new multi-resolution classification model called RANet, aiming for speeding up image classification under the assumption that some easy images can be classified at the lower resolution. The submission has an implementation of the RANet model in Tensorflow (with Keras), and the code is open sourced. This enabled Tensorflow users to try out RANet in the future, since the original authors implemented RANet in PyTorch.

2 main assumptions from the original paper are verified. The first is the existence of such distribution from easy- to hard-to-classify images, which is verified by showing that different subnets have different classification errors. The second is that RANet have an effective reduction in computational cost for image classification, which is verified using a classification task and a new earth quake detection dataset.

Being a good complementary study already, the submission could offer improvements in the following ways: 1) the submission should discuss how easy it is to reproduce the exact results in the original paper using the original authors' PyTorch implementation. It will form a better story for the current submission. 2) The authors should remove affiliation information on the computational resources used. This may run into anonymity troubles for more serious venues.

**Familiar With The Original Paper:**

I have read the original paper

**Reproducibility Summary:**

Report has summary

---

### Official Review · AnonReviewer3 · 2021-03-03
**Reproduction of smallest-scale results with extension to new dataset**

**Rating:** 5
**Confidence:** 4

**Review:**

**Strengths:**

1. Illustration of performance during training.

Table 1 is a nice result for reproduction purposes. The training dynamics of networks with conditional/adaptive components are known to be unstable sometimes. This reproduction shows, independently, how different subnetworks train within a CIFAR-10 experiment.

2. Considers an additional "spalling" dataset not in the main paper.

While not central to reproducing the claims in the original paper, it is good to investigate whether a method works on new data. One generally hopes that AI & CV can generalize beyond the standard datasets such as ImageNet and COCO, and the concrete spalling dataset in the reproduction has some major difference in the domain considered.

3. Full re-implementation of the RANet architecture.

The experiments in the reproduction were done in Keras, while the original RANet implementation is in PyTorch. Since there are still small differences in behavior between the two, it is good to know that the RANet architecture is not so implementation-dependent that it doesn't generalize beyond this.

**Weaknesses:**

4. Only reproduces some of the smaller-scale experiments.

The only dataset used in both the original paper and this reproduction is CIFAR-10. It is reasonable to cite computational requirements as a barrier to reproducing the experiments. However, it is a particularly major limitation when reproducing RANet, as the purpose of RANet is to adaptively reduce the resolution at which inference is done. CIFAR-10 images are already very low-resolution (32x32), having been selected from TinyImages.

If I understand correctly, the architecture in RANet is also adaptive w.r.t. to the number of layers processed (so some input-dependent adaptive behavior can be observed with this too), but this is also in previous work such as Veit & Belongie. The central new claim in RANet is in the handling of scale/resolution, that is better illustrated by ImageNet experiments (such as Fig 6c of Yang et al).

5. Some lack of clarity in dataset construction.

For the spalling dataset: were the images, or the patches, split between train and test? It is unclear to me based on my reading (around line 82). In Fig 3, it looks like some of the patches can overlap? Is it possible in the dataset contruction that there could be two overlapping patches, from the same image, for which one patch is in the training set and the other patch in the testing set? It would be less than ideal for train and test to be correlated in this way, due to having examples that share pixels.

**Misc Comments:**

  * 46: Typo "Tensoflow"
  * 46: "train own" -> "train our own"
  * 47: Is "be" meant to be "by"
  * 47: Should capitalize ImageNet

**Familiar With The Original Paper:**

I have read the original paper

**Reproducibility Summary:**

Report has summary

---

### Decision · Program_Chairs · 2021-03-31

**Decision:**

Reject

**Comment:**

Overall reviews and/or the paper content not good enough for the AC to recommend to the journal.